

# Chinese Interpreting Studies: a data-driven analysis of a dynamic field of enquiry

Ziyun Xu[1] and Leonid Pekelis[2]

[1] Intercultural Studies Group, Universitat Rovira i Virgili, Spain
[2] Department of Statistics, Stanford University, USA

## ABSTRACT

Over the five decades since its beginnings, Chinese Interpreting Studies (CIS) has evolved into a dynamic field of academic enquiry with more than 3,500 scholars and 4,200 publications. Using quantitative and qualitative analysis, this scientometric study delves deep into CIS citation data to examine some of the noteworthy trends and patterns of behavior in the field: how can the field's progress be quantified by means of citation analysis? Do its authors tend repeatedly to cite 'classic' papers or are they more drawn to their colleagues' latest research? What different effects does the choice of empirical vs. theoretical research have on the use of citations in the various research brackets? The findings show that the field is steadily moving forward with new papers continuously being cited, although a number of influential papers stand out, having received a stream of citations in all the years examined. CIS scholars also have a tendency to cite much older English than Chinese publications across all document types, and empirical research has the greatest influence on the citation behavior of doctoral scholars, while theoretical studies have the largest impact on that of article authors. The goal of this study is to demonstrate the merits of blending quantitative and qualitative analyses to uncover hidden trends.

Corresponding author
Ziyun Xu, xuziyun@gmail.com

## INTRODUCTION

There are various channels through which scholars communicate with one another, easing the flow of knowledge and furthering the advance of science. One such important channel comes in the form of citations, which are the result of the duty incumbent upon all scholars to conduct comprehensive and critical reviews of existing literature before embarking on new research, to gain a deep understanding of the field and find the precise empty niche into which their own work will fit, referring to previous related work to bolster their arguments. Though citing other people's work did not become the norm in scientific writing until the early 1900s (*Garfield, 1979*), it is now standard and required practice for authors to acknowledge the works of predecessors from which they have drawn inspiration, thereby maintaining the 'intellectual lineage' from one generation of academics to the next. Citation analysis has long attracted attention in the scientific community (see for example *Garfield, 1972*; *White & Mccain, 1998*; *Baumgartner & Pieters, 2003*; *Vallmitjana & Sabaté, 2008*). This is mostly as a consequence of

*Kuhn*'s (*1970*) ground-breaking work on the nature of science, in which he called on future scholars to recognise the crucial importance of adopting an empirical approach to studying the structure of a scientific community.

Such academic pursuits are particularly relevant in the Translation and Interpreting Studies (TIS) community, because it has experienced a significant growth in both quantitative and qualitative terms over the past two decades, and because hundreds of papers with diverse research methodologies and themes are produced on a yearly basis (*Franco Aixelá, 2013*). During this period of significant growth, more empirical studies are needed if we are to fully appreciate the patterns of communication and trends in TIS. A number of earlier scholars have used citation data to trace the evolution of the field and understand how scholars communicate with each other (see for example, *Pöchhacker, 1995*; *Gile, 2005*; *Grbić & Pöllabauer, 2009*). However, despite its usefulness, there are limitations to a purely quantitative approach in analysing TIS citation data, and qualitative analysis is called for in order to obtain a fuller picture of the discipline (*Gile, 2000*). The purpose of this scientometric study is to marry quantitative and qualitative approaches to analysing citation in order to obtain a panorama of CIS' evolution and reveal its hidden trends and predominant theoretical influences.

## BACKGROUND

### Major questions

CIS has been developing rapidly since the 1990s, as evidenced by its increasing number of publications and researchers (*Chen, 2009*). Using an all-but-exhaustive collection of citation data, three component strands of CIS (journal articles, MA theses, and doctoral dissertations) were studied with the aim of finding changes or differences in patterns of citation. In what ways is the citation network changing? Are authors still primarily influenced by older works or do more recent ones now hold the ascendancy? How do different research methods (theoretical, empirical, etc.) affect the use of citations in the works themselves? The three bodies of literature are generally produced by three distinct groups of authors: established researchers for journal articles and conference proceedings; graduate students for MA theses; and PhD students for doctoral dissertations (*Xu, 2014*; *Xu, 2015a*; *Xu, 2015b*). Examining these three strands individually is necessary if we are to fully understand how each contributes to advancing the field as a whole. Building on earlier studies by the present authors (*Xu, 2014*; *Xu, 2015a*; *Xu, 2015b*; *Xu & Pekelis, 2015*), which provided an overview of the field from different perspectives, this study uses some of the most sophisticated data-mining techniques currently available to answer the aforementioned complex questions, none of which can be adequately addressed by simple descriptive statistics.

### Literature review

The study of research trends in Translation and Interpreting Studies (TIS) is currently dominated by citation analysis (see for example *Gile, 2005*; *Gile, 2006*; *Gao, 2008*; *Franco Aixelá, 2004*). There are various methods of carrying out citation analysis, but the overall

basic concept is always the same. First, a sample of articles is selected; the researcher then counts the number of times each article is cited in other works. Citing (or 'source') works can be categorized according to type (conference proceedings, monographs, periodicals, etc.), and a weight assigned to each citation based on various factors: the type of publication in which it is being cited; the number of authors being cited; in the case of co-authorships an author's contribution to the work being cited (the 'target'); and others. Finally, a numerical score is calculated for each author, article, research institution, journal or whatever the researcher is focusing on; these scores can then be ranked to indicate each cited individual's or entity's relative impact (*Lowry, Karuga & Richardson, 2007*). The procedure is based on the premise that the number of times a work is cited is a measure of its influence in the academic world.

Citation analysis has increasingly been adopted to map out the historical evolution of a particular area of study, the impact of individual researchers, academic institutions or scientific publications, the extent of collaboration between these, or the influence of certain disciplines on others (*Glänzel, 2003*; *Kalaitzidakis, Mamuneas & Stengos, 2003*). In their general study of the technique, *Braun, Glänzel & Schubert (1985)* found that articles cited between five and ten times each year during the period immediately following their publication tend to be assimilated into the relevant discipline's 'universal' stock of knowledge, and that conversely, if articles go uncited over the same period, there is little chance of such assimilation taking place. Citation analysis has been used in well-established disciplines such as linguistics (*White, 2004*), psychology (*Carr & Britton, 2003*; *White & White, 1977*), and information science (*White & Mccain, 1998*), but has also been highly useful in assessing research patterns in fields with much shorter histories, such as TIS (*Gile, 2005*).

Given the increasing popularity of citation analysis, Garfield's Institute of Scientific Information (ISI) produced the first citation index[1] for articles published in academic journals shortly after it was founded in 1960. The ISI has since produced numerous other indexes, which have grown to encompass more than 40 million records and 8,700 research journals (*Meho, 2006*) and are now accessible online via Thomson Reuters' Web of Science. Although originally designed to facilitate access to information, the indexes are now widely recognized as an important source of empirical data for scientometric research (*Ivancheva, 2008*).

Despite the growth in use of citation indexes, the exponential expansion of scientific research into new disciplines over the past four decades has resulted in numerous high-quality journals being excluded from the 'baskets' used by the leading indexes. To facilitate improved communication among researchers in the field of interpreting, in 1990 Daniel Gile set out to create an international network—the Conference Interpreting Research Information Network (CIRIN)—which publishes a biannual Bulletin. Since then several other searchable databases have been created for this discipline: the Bibliography of Interpreting and Translation (BITRA), for example, carries over 50,000 entries and is updated on a monthly basis, while the Translation Studies Bibliography (TSB) subscription service has 24,500 entries to date.

[1] A citation index is a database that archives bibliographic information from publications: it allows users to trace the progress of a concept or subject of inquiry by sourcing published works that cite particular authors or articles.
*Gile (2005)* surveyed citations from 47 papers on translator and interpreter training written by Western academics to find out which theories were most influential, the languages that target works were most often written in, and whether empirical or non-empirical research had more influence. The interpreter training material he sampled for the study revealed several interesting points: the model advocated by the Association Internationale des Interprètes de Conférence (AIIC) was the most frequently cited theory, while functional theories were dominant in translator training; the majority of the cited literature was written in English; and empirical research played very little part in the papers sampled. In another study (2006) he introduced a qualitative dimension to his analysis by grouping citations into different categories (concepts, methods, findings, etc.), on the assumption that such an approach would provide a more nuanced analysis of each category's impact on the evolution of Translation and Interpreting Studies (TIS). The study revealed that scholars were cited on their methods and findings in less than 10% of the articles in the corpus. Adopting the same classification scheme, *Nasr (2010)* examined a corpus of 542 texts on translator training. Her study produced a similar result, indicating that empirical research was not influential in shaping research into that subject either.

By developing methodologies based on citation analysis, earlier researchers have laid the groundwork for assessing the impact of an individual's work and tracing the evolution of a field. In addition to quantitative analysis, qualitative approaches have been proposed to study how scholars cite one another. However, the application of these methodological techniques to investigating the evolution of CIS has to date been very limited. The goal of the present study was to adopt a blended approach with equal emphasis on both quantitative and qualitative considerations to explore how the CIS citation network changes over time and how different research methodologies have affected citation behaviours.

## THE PRESENT STUDY

Expanding on the broad themes of enquiry outlined at the beginning of this paper, three more in-depth mini-studies were drawn up to address some of the major issues unresolved by previous researchers. The rationale for each is summarized in the following section.

### Data organization

The authors created a near-comprehensive database of 59,303 citations from the 1,289 Chinese MA theses, 32 doctoral dissertations and 2,909 research papers available to them. The CIS literature was collected from multiple databases and other sources: field trips to university libraries, interlibrary loans, book purchases, and academic databases such as CNKI, Wanfang and the National Digital Library of Theses and Dissertations in Taiwan. Publications with no bibliographic references were excluded from the analysis. Every effort was made to ensure that the field was covered as widely and exhaustively as possible. Though the contents of a handful of embargoed theses were inaccessible, the present study authors analysed those of their features that were available, such as titles, publication years and abstracts. Essentially the entire population of CIS studies was sampled, making it possible to generalize the conclusions of this study to the entire field. Once collected, the
references were manually entered into a relational database which uses Structured Query Language (SQL) for managing data.

## Study 1

### Research question

Do CIS authors tend repeatedly to cite 'classic' papers, or are they more drawn to the latest research within the field? How can the progress of CIS be quantified by means of citation analysis?

A number of scholars (*Merton, 1967*; *Lederberg, 1972*; *Garfield, 1977*) have observed that at the same time as science constantly moves forward, there exists a phenomenon known as obliteration: the pace of scientific progress is so rapid, and new findings become so quickly and thoroughly absorbed into the 'general stock' of knowledge, that a great deal of work is quickly 'forgotten' by the academic community. The phenomenon is particularly noticeable in exact sciences, in which authors seem consistently to build upon relatively recent research, the time lag between an author and the work he cites remaining fairly constant (*Van Raan, 2010*). At the same time, other scholars have observed the long-lasting impact of 'classic' works on the evolution of a field. For instance, *Franco Aixelá*'s *2013* study of the most cited works in Western Translation Studies (WTS) revealed that almost all the most frequently cited papers were "classics" published well before the 2000s, a finding which appears to suggest that WTS scholars have a marked preference for deepening and widening their understanding of the ages-old issues of translation and otherwise carrying on the intellectual lineage of classical authors.

The research by the aforementioned authors points to two contrasting patterns of knowledge flow existing in tandem. *Merton (1967)*, *Lederberg (1972)*, *Garfield (1977)* and *Van Raan (2010)* identified a scenario whereby knowledge flows at a steady rate, referred to in the remainder of this chapter as 'perfect research flow'. By contrast, *Franco Aixelá (2013)* has observed deviations from this scenario, proposing a continuum of flow rates reaching to the extreme opposite of 'research stagnation'. The aim in this section was to discover whether or not the CIS community followed this academic tradition of WTS' and, more generally, to examine how the field's progress could be illustrated by means of citation analysis.

### Research methodology

Two null hypotheses were tested: the first was that of 'research stagnation'[2]—this tests whether new papers are not constantly being cited; and the second was that of 'perfect research flow'—this tests whether the citation process is stationary.[3]

*The hypothesis of research stagnation.* Research stagnation occurs when articles published after a given year ($t$) suddenly cease completely to be cited. One scenario which can lead to this state of affairs is when articles published before year $t$ are so influential that they 'drown out' all citations from ones published after it. This hypothesis is rejected if new papers are being constantly cited.

[2] 'Research stagnation' in this context is shorthand for the stagnation of the citation process, whereby new articles are not cited and therefore after a given year the distribution of citations falls to zero. As we shall see in the Results and Discussions, this hypothesis was later rejected. Of course, numerous other factors need to be taken into consideration to determine whether or not a field of inquiry is moving forward. Unfortunately the analysis of these is outside the scope of the present study.

[3] A process is said to be stationary if its distribution remains unchanged over time. In the example given in The Hypothesis of Perfect Research Flow the distribution of papers cited in year $t$ is said to be stationary if its relation to the previous years ($t$, $t-1$, $t-2$, etc.) does not depend on $t$.

*The hypothesis of perfect research flow.* Perfect research flow occurs when the citation process is stationary. The following example illustrates a case of perfect research flow: for articles published in a given year $t$, let us suppose that no citations come from year $t - 4$ or earlier, and that most citations come from papers published in year $t - 3$, with half as many for each successive year down to $t$ itself. Perfect research flow comes about when this distribution of citations is true for all the years $t$ examined in the study.

A typical scenario that would cause this hypothesis to be rejected would be if a few very influential ('classic') articles were published in a given year $t_0$ and cited more than the average article, even ten years later: in this case the citation process would indeed not be stationary, because in year $t_0 + 10$ citations of this article published ten years previously would still be being produced! We therefore would not be dealing with a case of perfect research flow.

*Hypothesis testing.* The aforementioned hypotheses concern the distribution of the citation process. To test them, all the papers published in year $t$ and the years of all citations contained in those papers were identified. The distribution of papers cited in year $t$ was estimated as the average number of citations per paper published in year $t$ coming from each previous year: $t - 1$, $t - 2$, $t - 3$ and so on. The same methodology was applied to all publication years between 1990 and 2013. Once the distribution of cited papers for each year $t$ was established, it was possible to test whether the figure was stagnant, and, by measuring how it changed from year to year, whether it was stationary.

These two hypotheses were tested by comparing the performances of two models—one for each hypothesis—to that of a third, namely a varying coefficient model (VCM).

VCMs are more generalized versions of regression. Regression expresses the value of an output as a combination of different types of input (or predictors). Each input has an associated coefficient which signifies the importance of its contribution to explain the output. In varying coefficient models, the coefficients themselves vary with other variables, which may or may not be connected to the predictors. For example, in the context of a chemistry experiment, we may get very different coefficients in a linear regression of amount of reagent created depending on outside parameters such as temperature. A VCM would give a better fit: the coefficients in the regression are functions of the temperature (this is not the same as including the temperature as a predictor, since the dependence of reagent created to the temperature is not direct or linear). An overview of the theory behind VCM models can be found in *Hastie & Tibshirani (1993)*, and *Fan & Zhang (2008)* contains an excellent review of the many ways in which such models are applied and implemented.

We used a Varying Coefficient Model to fit a citation process somewhere between research stagnation and perfect research flow in the following way. In each model the output was the average number of citations per paper published in target year $t$. For research stagnation the input was the raw source year of the citations (e.g., a paper is published in 1996). For perfect research flow the input was the relative source year of the citations (the relative source year of papers published in year $t - i$ is $i$). For both of these

**Table 1 Evaluation of VCMs.**

| | Research stagnation deviance | Perfect research flow deviance | VCM model deviance | P-value VCM < RS | P-value VCM < PRF |
|---|---|---|---|---|---|
| MA | 341.9 | 845.40 | 247.22 | <0.001 | <0.001 |
| Journal | 3,871.26 | 10,124.87 | 1,909.24 | <0.001 | <0.001 |
| PhD | 67.0 | 80.45 | 29.25 | <0.001 | <0.001 |

models the input coefficients were forced to be fixed across target year $t$. Finally, we fit the third—VCM—model with the same inputs as for perfect research flow, but allowed the coefficients to vary smoothly with source year.

All three models were fit using a generalized linear model with Poisson link function. Additionally, the VCM was fit using locally weighted least squares and a Gaussian kernel.

To test the null hypotheses of research stagnation and perfect research flow we examined whether the VCM model fit the data significantly better than either of the first two using a generalized deviance difference test as proposed in *Fan, Zhang & Zhang (2001)*. Since the technical details and assumptions of such tests often depend in complex ways on particular features of the data-set, a bootstrap procedure was used to calculate $p$-values non-parametrically. In other words, the $p$-values for this analysis are adaptive and automatically fair to the features of the current data-set. For a general description of hypothesis testing with the bootstrap, see *Efron & Tibshirani (1994)*, in particular Chapter 16. Table 1 below contains the resulting $p$-values.

A more detailed description of our statistical methods—model description, fitting procedure and hypothesis tests—can be found online at: http://interpretrainer.com/VCM_Justification.pdf.

### Results and discussions

Figure 1 represents the distribution of citation processes for MA theses, doctoral dissertations and journal articles in different years.

*Hypothesis of research stagnation.* Figure 1 indicates definite movement over time for the incoming citation curves. If all the curves in panel 4 of the figure had looked the same, this would have supported the hypothesis that the field of CIS is static. This is not the case here: the 'peaks' in the curves move forward from year to year and do not 'stagnate' at a given year. In sum, the figures suggest that CIS research is moving forward.

In addition, the hypothesis of research stagnation was rejected on statistical grounds: more recent CIS publications were constantly being cited, as opposed to classic papers receiving the majority of citations as time went by, and that caused the model corresponding to research stagnation to fit less accurately the data than did the VCM model, as demonstrated by the very low $p$-values for the corresponding tests[4] (see Table 1 for more information).

While newer citations may not necessarily contain innovation—instead simply restating the positions found in classic works—there is assuredly some foundation to Zuckerman's

[4] 'The corresponding tests' refers to those that compare the research stagnation model against the VCM model to see if the former fit data better than the latter. This hypothesis was rejected for all document types: MA theses, journal articles and PhD dissertations.

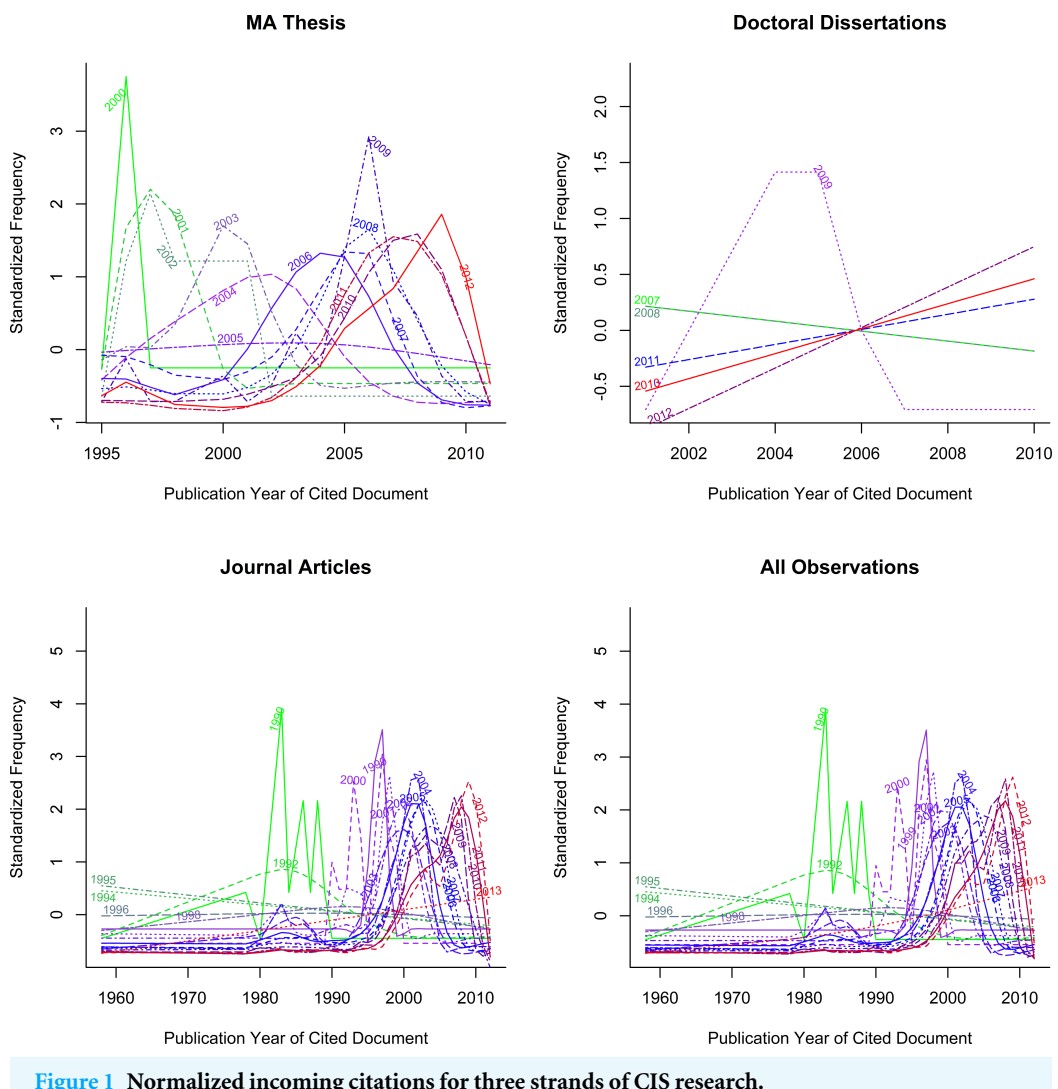

**Figure 1 Normalized incoming citations for three strands of CIS research.**

argument (*1987*) that the use of more recent citations nonetheless indicates that academic inquiry is moving forward. The argument is as follows: a cited paper (Paper A) gains influence when it is cited by multiple authors; however, authors may sometimes be inclined to cite other more recent papers that specifically refer to Paper A, as opposed to citing it directly. While these more recent publications may or may not generate new findings or innovative material, they effectively serve as an intellectual conduit connecting contemporary researchers with past foundational work. Paper A has become so thoroughly incorporated into the field's stock of knowledge, has become so fundamental to it, that authors feel no need to make explicit reference to it. Therefore, the rejection of the research stagnation hypothesis indicates that contemporary researchers build on more recent work and that academic enquiry is moving forward.

Even though it was both visually and statistically confirmed that CIS is moving forward, whether it has been doing so at a steady pace remains an open question. Rejection of the
research stagnation hypothesis tells us nothing about how research evolves, in particular it sheds no light on the question of whether the flow of research is 'perfect' in the sense that the distribution of citations remains the same from year to year. Hence the need to test the hypothesis of perfect research flow.

*Hypothesis of perfect research flow.* Perfect research flow is the extreme opposite of stagnation; it means that papers are cited in exactly the same fashion every year. Figure 1 also enables us to grasp visually the rejection of the stationarity hypothesis. If this hypothesis were true, it would imply that the lines shown in the plots did not change with source year. This is clearly not the case here.

Additional statistical analysis was conducted to confirm the visually striking evidence in Fig. 1 regarding the hypothesis of perfect research flow. Indeed, this hypothesis was rejected on statistical grounds, because the Varying Coefficient Model fit better to the citation data than did the model corresponding to perfect research flow.[5] Once again this rejection is demonstrated by the very low $p$-values of the corresponding tests in Table 1.

[5] This rejection is described in detail at http://interpretrainer.com/VCM_Justification.pdf.

*Hypothesis testing and graphical interpretations.* To test both of the previously mentioned hypotheses a VCM model was used first to describe the data as accurately as possible, then this model's performance was tested to compare it with those of the models corresponding to each hypothesis.

For each year $t$, a spline was fit to incoming citations as a function of $|t - i|$, where $i$ was the year of publication of the cited article. The VCM model was constructed so that it would be easy to control the variation of the coefficients over time.

The resulting graphs (see Figs. 2–4) can be likened to a frame-by-frame film of the evolution of incoming citations over time.

The red line is the fit for the VCM model and can be considered the average citations count for that year; the blue dots are the actual number of citations produced in each year; and the grey shaded areas represent a 95% confidence interval for the red line. The grey headers show the year under consideration—for example, '2000' means that all the papers written in 2000 were examined to ascertain the number of citations in them dating from 2000 ($t$), 1999 ($t - 1$), 1998 ($t - 2$), and so forth.

Examination of the incoming citation data revealed that recent papers were regularly cited within an interval of a year or two—this trend was particularly obvious from 2009 to 2012. *Moed (2005)* has argued that an author might include a certain reference not only because its content fits the flow of an argument, but because he believes the scholar he is citing has gained a certain stature in the field and will lend credibility to his own ideas. For example, it would be more credible to cite a definition of empirical research formulated by a scholar who has conducted extensive studies of that type than by one whose focus is purely theoretical. The finding that recent papers are cited so soon indicates that newer research has a more or less instant impact on the latest studies and that CIS research is in a state of continuous progression. It was also remarked that, in disregard of the 1–2 year rule mentioned above, citations from material published in 1990 were made in CIS papers throughout the period under study, suggesting that that year may have seen

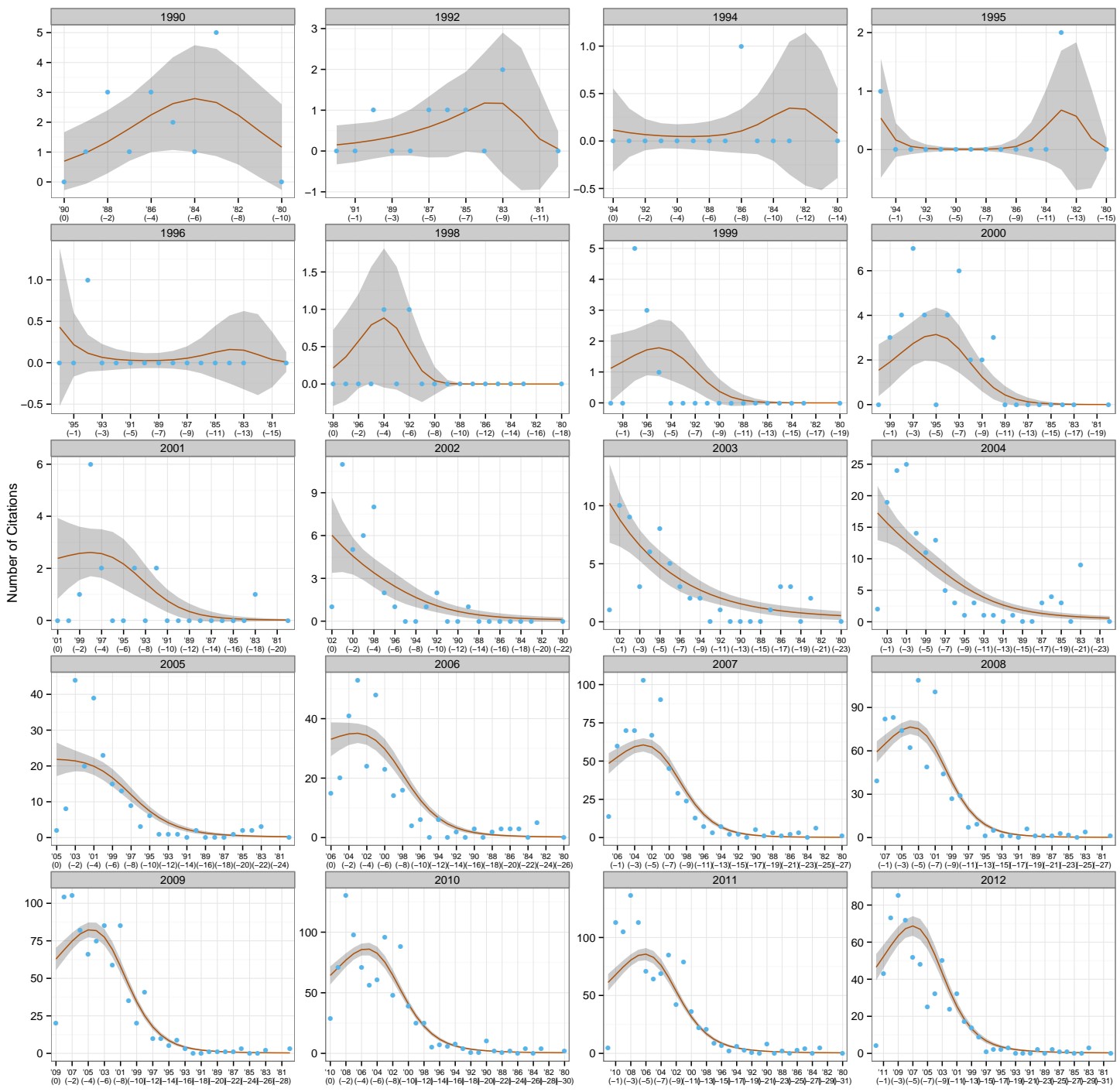

**Figure 2  Trends in citations for research papers.**

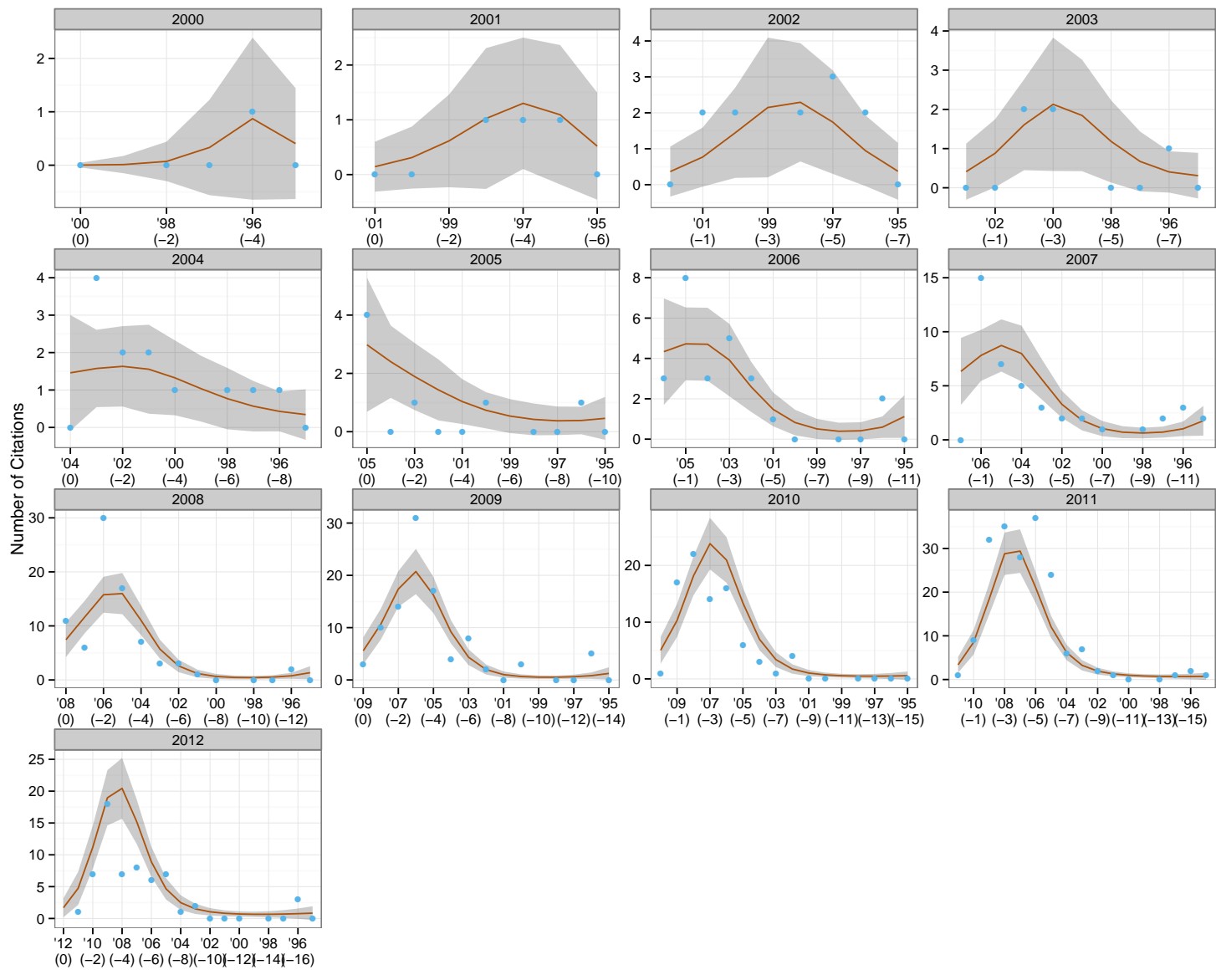

**Figure 3  Trends in citations for MA theses.**

the publication of particularly influential material, whose impact on research has been especially long-lasting. On further examination of the incoming citations, Hu Gengshen's (*1990*) *An overview of interpreting research in China* stood out as the aforementioned material. Hu's paper took a scientometric approach to assessing the themes and trends in interpreting research. From the Y axis it was also clear that many more citations were being made in later years, probably because the number of CIS papers being written was increasing year on year.

The situation for MA theses was slightly different from that of journal articles, though research was moving forward here too. These authors were somewhat hesitant to cite recently completed theses, preferring those produced at least three years previously, which

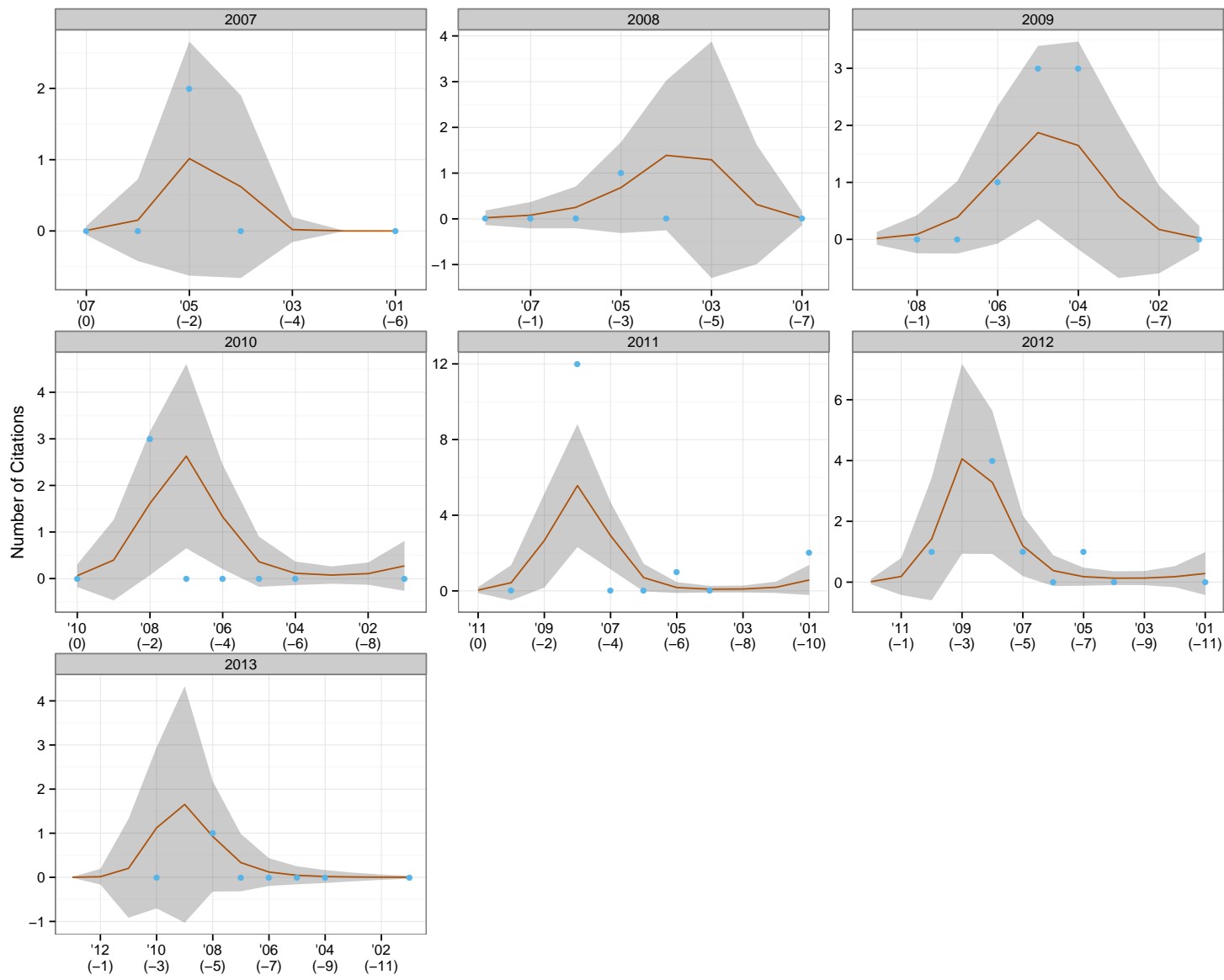

**Figure 4  Trends in citations for doctoral dissertations.**

they could be sure, had been adopted by the academic community and become established. It was also noticeable that material produced in 1996 was cited by numerous MA authors in all subsequent years, suggesting that some very influential work was produced in that year. Detailed analysis revealed that work to be Ru Mingli's (*1996*) thesis *Interpreting quality and the role of the interpreter from the perspective of users*, which was produced under the supervision of Chen Yongyu. It should be noted, however, that MA thesis authors cited their predecessors' work far less often than the authors of research papers did theirs: in 2010, for example, research papers produced in 2008 were cited no fewer than 148 times; the same figure for MA theses was a mere 22. There are two possible reasons for this phenomenon. Firstly, a number of researchers (*Lawrence, 2001*; *Harnad & Brody,*

*2004*; *Hajjem, Harnad & Gingras, 2005*) have identified that open-access articles receive a substantially higher number of citations than those that require a subscription—this is true across many disciplines including computer science, physics, sociology and psychology. Proceeding from their findings, it is reasonable to speculate that the difficulty—and expense—of obtaining access has contributed to the significantly lower number of CIS theses being cited in comparison to research papers. Secondly, in the academic world MA theses are generally considered to be of lower quality than research papers, which have gone through rigorous peer review.

Given that the total number of doctoral dissertations was only 32, little in the way of trends was observable. It should be noted, however, that a particular doctoral dissertation produced in 2008 was consistently quoted by later PhD authors in the period 2010–13—this was Gong Longsheng's (*2008*) *An analytical study of the application of Adaptation Theory in interpreting*, written under the supervision of Dai Weidong. Gong is such a well-established and visible academic within the CIS community,[6] that it is hardly surprising that his work might attract a large number of incoming citations.

To conclude, two null hypotheses were both visually and statistically rejected: research stagnation and perfect research flow. To perform those tests two models corresponding to each of the hypotheses, and a third, the Varying Coefficient Model, were constructed. The three were tested to see how well they fit the CIS citation data. Both hypotheses were rejected, because the first two models performed poorly in comparison with the VCM model. Analysis of the citations yielded enough evidence to say that this field is going forward, though not at a uniform pace.

## Study 2

### *Research question*

**What are the most frequent citation types? Do they differ based on language of origin (Chinese vs. English) and document type (papers, MA theses and doctoral dissertations)?**

While quantitative analysis of the academic influence of individual authors, institutions, geographical regions and publications may shed light on the entire CIS landscape, some qualitative analysis of citation types is necessary to provide insight into the interactions between the various schools of thought and research practice that constitute the field.

### *Research methodology*

The citations were labelled according to the way each cited paper was used by the referring paper. After an initial pilot study,[7] a citation classification system was developed to evaluate how authors were cited in the CIS literature (see Table 2).

A random sampling without replacement was conducted on each of the six citation databases of CIS: English citations in MA theses, research papers, and doctoral dissertations; and Chinese citations *ibidem*. This form of sampling was used because it leads to more accurate results than sampling with replacement, thanks to an effect known in the simulation literature as 'variance reduction' (*Rao, 1963*). The minimum sample size was fixed at one large enough for detecting any statistically significant difference in the

[6] Gong served as an associate dean of the Graduate School of Business at SISU, has supervised nearly 30 MA interpreting students, and was involved in developing the Shanghai Interpretation Accreditation Test.

[7] Building on the studies of *Gile (2006)* and *Nasr (2010)*, the authors conducted a pilot study of citation types in CIS by randomly sampling a total of 239 in-text citations from Chinese papers, theses and doctoral dissertations. After labeling these according to the methods described in Gile's pilot research project (*2006*), a collaborator who was familiar with the topic was asked to give a 'second opinion' by labeling them again himself, with the aim of ensuring a greater measure of objectivity and reliability. These labelling activities were completed in four installments. The sequential analysis was restricted to 239 citations because by that point the author had sufficient knowledge of regularly occurring citation types in CIS, and the differences in labeling between the author and the collaborator were minimal.

**Table 2  Citation classification system.**

| Citation types | Definitions |
| --- | --- |
| Prescriptive opinion | The cited author explicitly expresses his position on an issue or issues, and directs readers. Statements of this nature often rely on modals such as *ought*, *should* and *must*. |
| Non-prescriptive opinion (Claim) | The cited author expresses a personal view but *without* directing the reader. |
| Assessment | The cited author gives an evaluation of an issue or issues he deems important. |
| Concept | The cited author puts forward a detailed idea. |
| Rules/standards | The cited author talks about principles of conduct or codified regulations. |
| Theory/model | The cited author creates a group of propositions that are used to explain or predict certain phenomena. |
| Theoretical analysis | The cited author examines a phenomenon, concept or behavior in abstract terms, basing his reasoning on existing theoretical frameworks. |
| Idea | The cited author's thought is non-technical and lacks the detail found in 'concepts', 'theories' and 'theoretical analysis'. |
| Tangential Research | The cited author has made a detailed study of a particular subject in the hope of obtaining new information or deepening understanding. |
| Research method | The cited author adopted a particular approach to uncovering new information or advancing understanding. |
| Research finding | The cited author draws factual or empirical findings from a study. |
| Non-TS factual citation | The cited author covers factual information outside the scope of Translation Studies. |
| TS-related factual citation | The cited author alludes to factual information that falls within the scope of Translation Studies. |
| General Principle | The cited author talks of fundamental 'truths' which fall short of being absolute. |
| General Report | The cited author observes and describes a phenomenon or behavior. |
| Qualification | The cited author sets parameters or defines the limitations of a phenomenon. |
| Definition | The cited author explains a phenomenon, behavior or concept in terms of its unique characteristics. |
| Characterization | The cited author describes the features of a particular issue or concept. |
| Textual | The citation is of an author's textbook or other instruction manual. |
| Non-theoretical analysis | The cited author makes a face-value examination of certain issues or phenomena without the use of theories. |
| Famous quotes | The cited author quotes a well-known person's words to emphasize a point. |
| Miscellaneous | Any citations that do not fit into the aforementioned categories. |
| Not available | A bibliographic reference does not correspond to any of the in-text citations in a paper. |

proportions of each citation type between Chinese and English. Given that we had no prior knowledge about whether the number of Chinese citations would be larger than that of the English ones, or vice versa, a power calculation for a two-sided two-sample proportion test was conducted for each of the six citation databases in order to give a 95% chance of detecting a difference of 10%[8] between two proportions at the 0.05 significance level. The minimum number of samples required for each database (648 as determined by the power calculation) was collected, analysed and labelled.

The majority of the in-text citations were expected to belong to a single citation type, but some belonged to more than one—this occurred primarily when an author cited a particular source at different places in his text. If one of these multi-category observations was selected for inclusion in the random sample, that citation needed to be assigned to a *single* category to maintain consistency. To address this, one of the assigned categories with equal probability was selected at random to decide which category the citation should have been assigned to. Though it may appear crude, assuming equally probable categories as a first

[8] If the true difference in proportions were, for example, 1%, no difference would be detectable unless the sample sizes were at least 64,974! The entire population for the present study comes nowhere near this size. A 10% difference was chosen because it allowed the author to work with a reasonably sized sample.
approach yields good results, and is frequently used (*Kempthorne, 1952*; *Freedman, 1997*; *Schulz & Grimes, 2002*) when there is no prior knowledge of the distribution of the data.

The results of the power calculation gave us an effective way of sampling; we were subsequently able to construct confidence intervals of 95% for the proportion of each citation type. It was expected that N/A would be the most frequently used citation type across all three categories of CIS publication in both Chinese and English. Citation practice in the Chinese academic community is distinctly different from that of the Western world: large numbers of scholars list in their bibliographies the literature they consult while conducting research, even if it is not directly cited in the body of their texts. However, according to most Western style guides, such as that of the American Psychological Association (APA), authors are required to cite the works of those who have directly influenced their research, and every resource in the bibliography must have a corresponding in-text citation, with the exception of some classics such as the Bible (*APA, 2010*). Because of the lack of textual references to these N/A citations it was not easy to code them. There may be limits to the space available to authors for recording references, meaning that not all the works that have influenced them will make it to the final list, so it is reasonable to assume that they are painstaking in their choice of what to include. Those citations that were listed, including those in the N/A category, must have had a major influence on the author, be it factual, theoretical or inspirational. Because of the different impacts that Western and Chinese literatures have had on CIS, the proportions of N/A for English and Chinese citations were expected to differ—analysing these proportional differences might help to illustrate the ways in which Eastern and Western thought have influenced CIS.

Another expected finding was that authors would be frequently cited for their Ideas and Prescriptive and Non-prescriptive Opinions; this may come as a surprise to those from disciplines where opinion-based citations are not common. Such citations are prevalent because practising interpreters are highly respected in the CIS community. This is corroborated by the fact that numerous professionals with no background in research are regularly invited to serve as keynote speakers at the biannual National Conference and International Forum on Interpreting, the most important research conference in China. Tangential Research was another citation type expected to be in frequent use, because scholars may often feel obligated to make 'ceremonial citations', i.e., referencing the leading experts in the field without actually having read their research (*Meho, 2006*). At the other end of the spectrum, Research Methodology and Findings were expected to be in much less frequent use, because non-empirical research still accounts for the overwhelming majority of published works in CIS.

### Results and discussions

When a fair and unbiased random sample is taken from a large population, the law of large numbers ensures that the sample average for any category ought to be close to the true value of the mean in the total population. Confidence intervals represent the estimated range within which the true average proportion of each category will probably fall; in the present study this probability was set to 95%. For example, in our sample (3,888 citations),

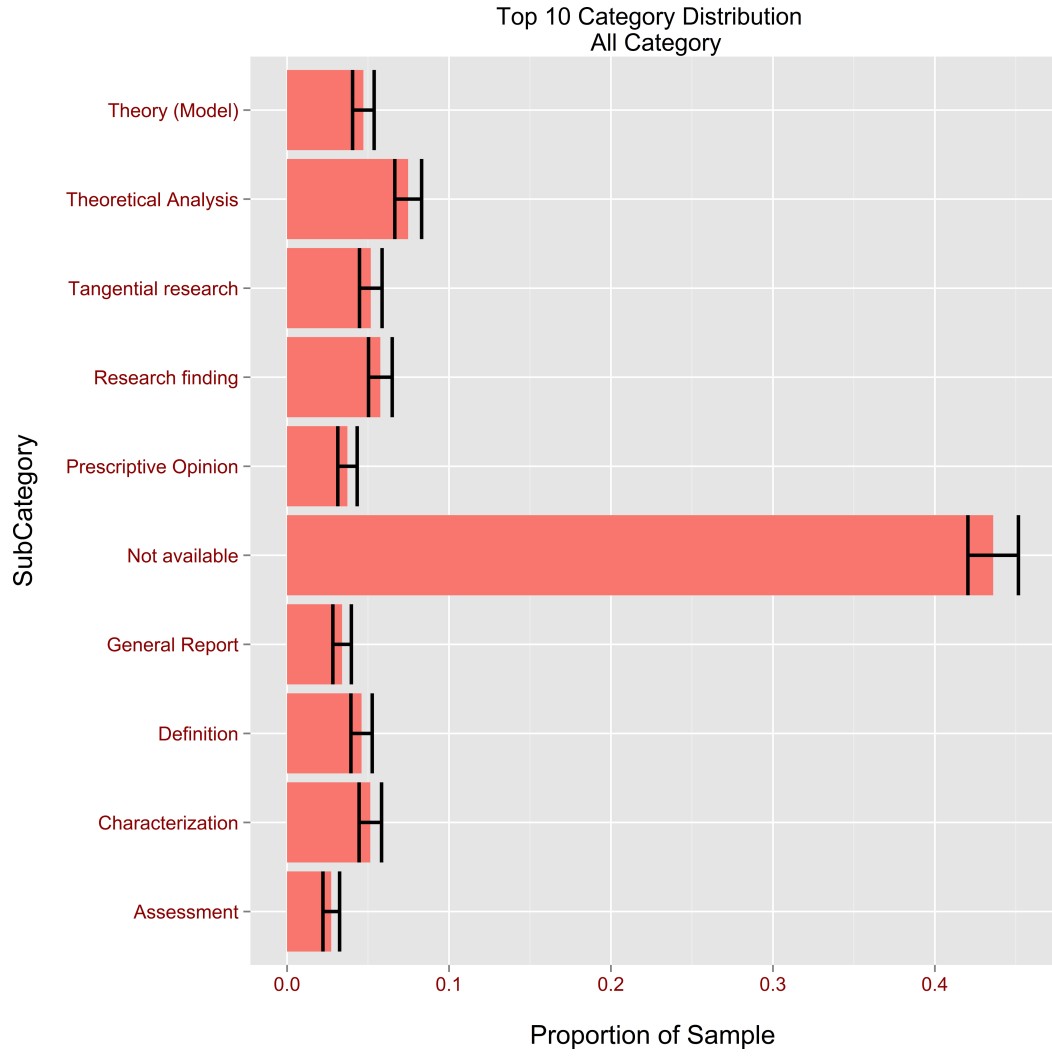

**Figure 5 Proportion of the top 10 citation categories with confidence bars across all databases.**

nearly 45% were labelled N/A (see Fig. 5). Therefore the 95% confidence interval for all the N/A citations was [42%; 45.1%], a finding which appears to suggest that almost half of the works listed in the bibliographies of CIS authors did not appear in the body of the text. The prevalence of N/A references indicates that certain cited authors may not necessarily be linked directly to the research of those citing them, despite having played an instrumental role in shaping their outlook on interpreting or influencing their professional training. For example, a further analysis revealed that 75.3% of the references to Mei Deming and 85.6% of those to Zhang Weiwei belonged to the N/A category; their cited works are the leading interpreting textbooks in China, though not regarded as theoretical or empirical contributions to CIS research. The second most popular citation type (Theoretical Analysis) stood at 7.5%; the corresponding 95% confidence interval was [6.6%; 8.3%]. From this finding it is reasonable to speculate that theoretical research has played a crucial role in shaping CIS. It was also observed that 5.8% of citations belonged
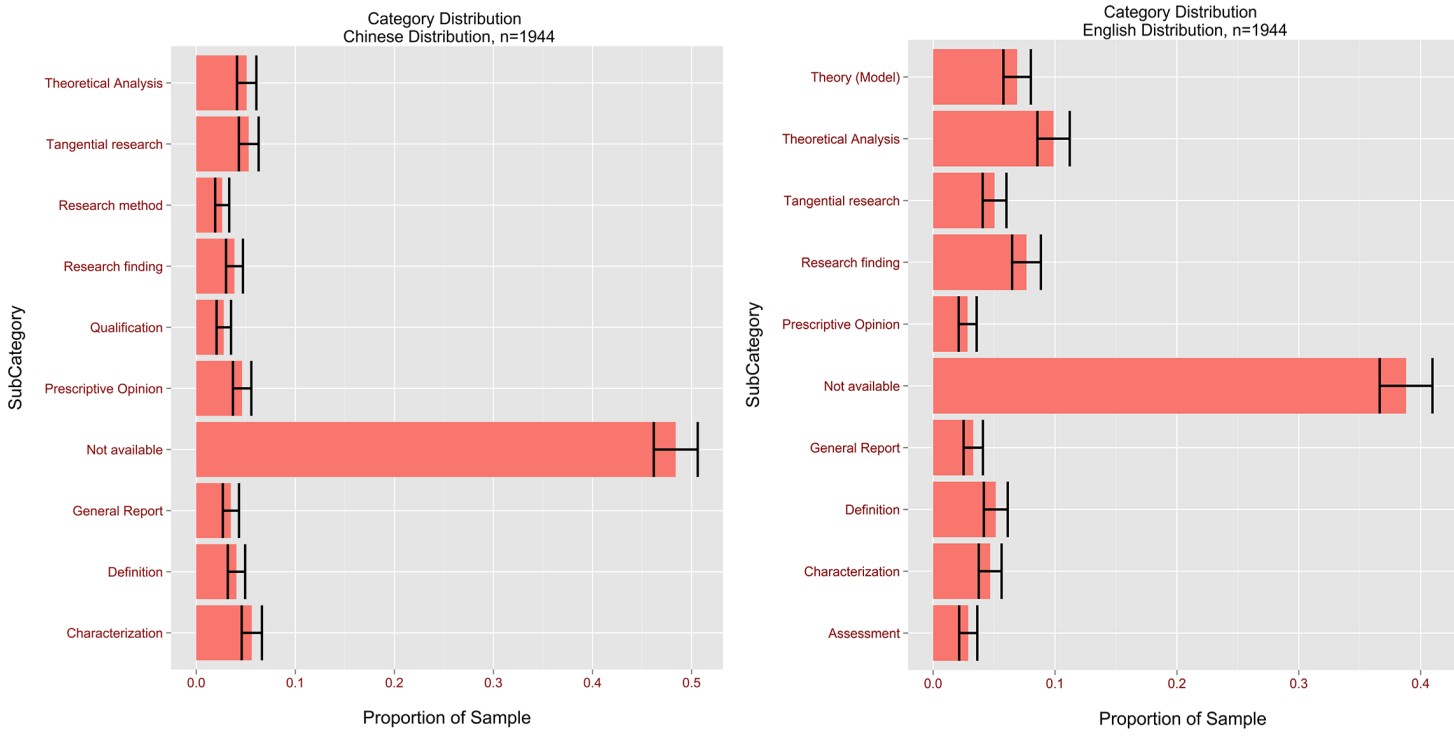

**Figure 6** Proportions of all English and Chinese citation type distributions with confidence bars.

to the Research Finding category (95% CI [5.1; 6.5%]). This is interesting because that citation type is generally associated with empirical research. Its being the third most frequently used type seems to indicate that CIS authors were also keenly aware of the importance of empirical research, and consciously analysed how their research could amplify the findings of previous scholars.

An examination was also made of the distribution of citation types by the following methods:

(1) Citation type distribution for all Chinese citations (see Fig. 6)
(2) Citation type distribution for all English citations *ibidem*
(3) Top ten citation types for all Chinese citations in theses, dissertations and papers (see Fig. 7)
(4) Top ten citation types for all English citations in theses, dissertations and papers *ibidem*
(5) Citation types for MA theses, dissertations and papers (see Fig. 8)

Confidence intervals of 95% were constructed to compare the differences in the proportions of each category between Chinese and English references. If for a given category the means of the two populations had non-overlapping confidence intervals, this indicated a statistically significant difference between the Chinese and English citations—non-overlapping 95% confidence intervals guarantee a test for differences at an alpha level of 0.05 (*Knezevic, 2008*). For example, the proportion of N/A citations

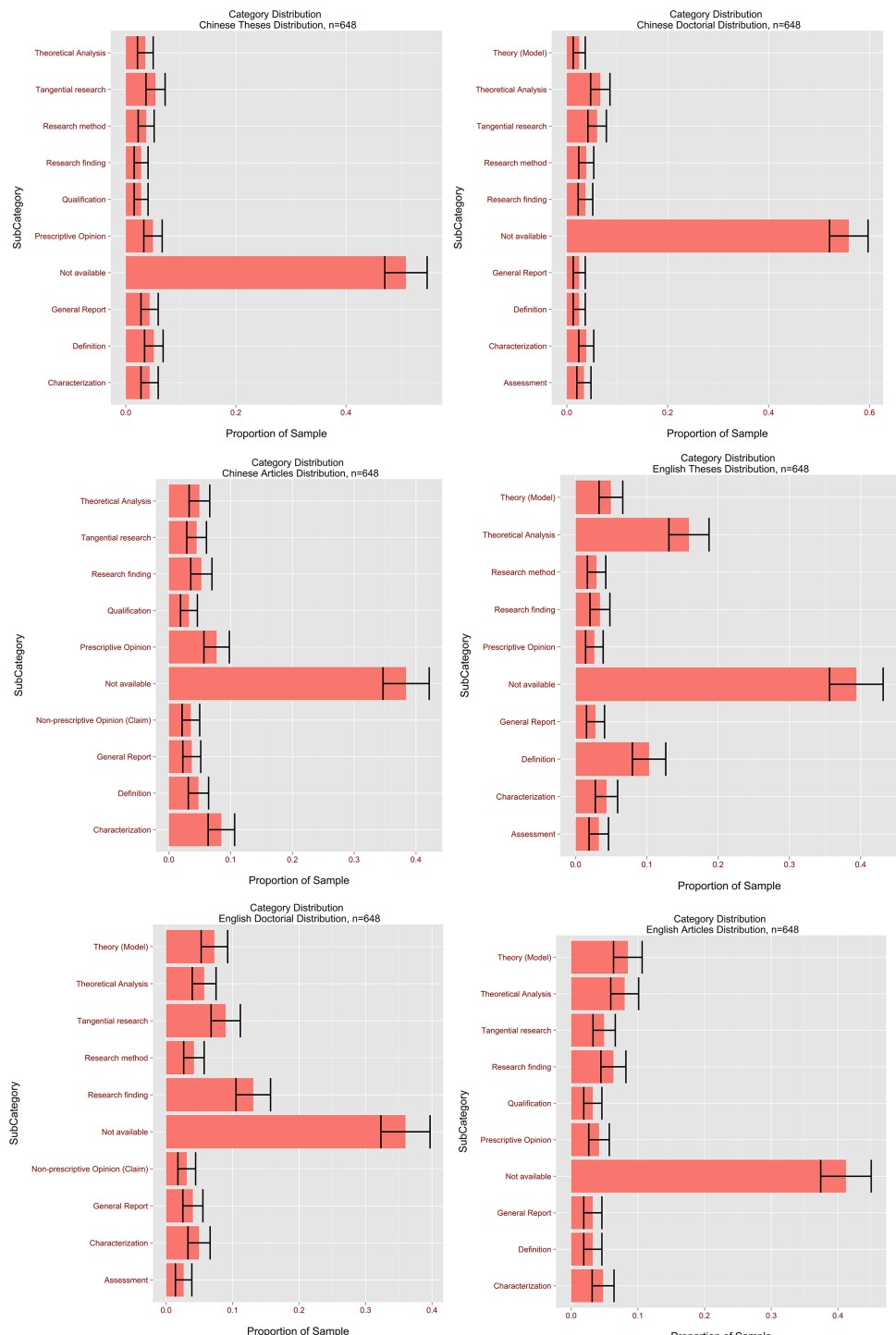

**Figure 7 Proportions of the top ten citation types for all Chinese and English citations in theses, dissertations and papers.**

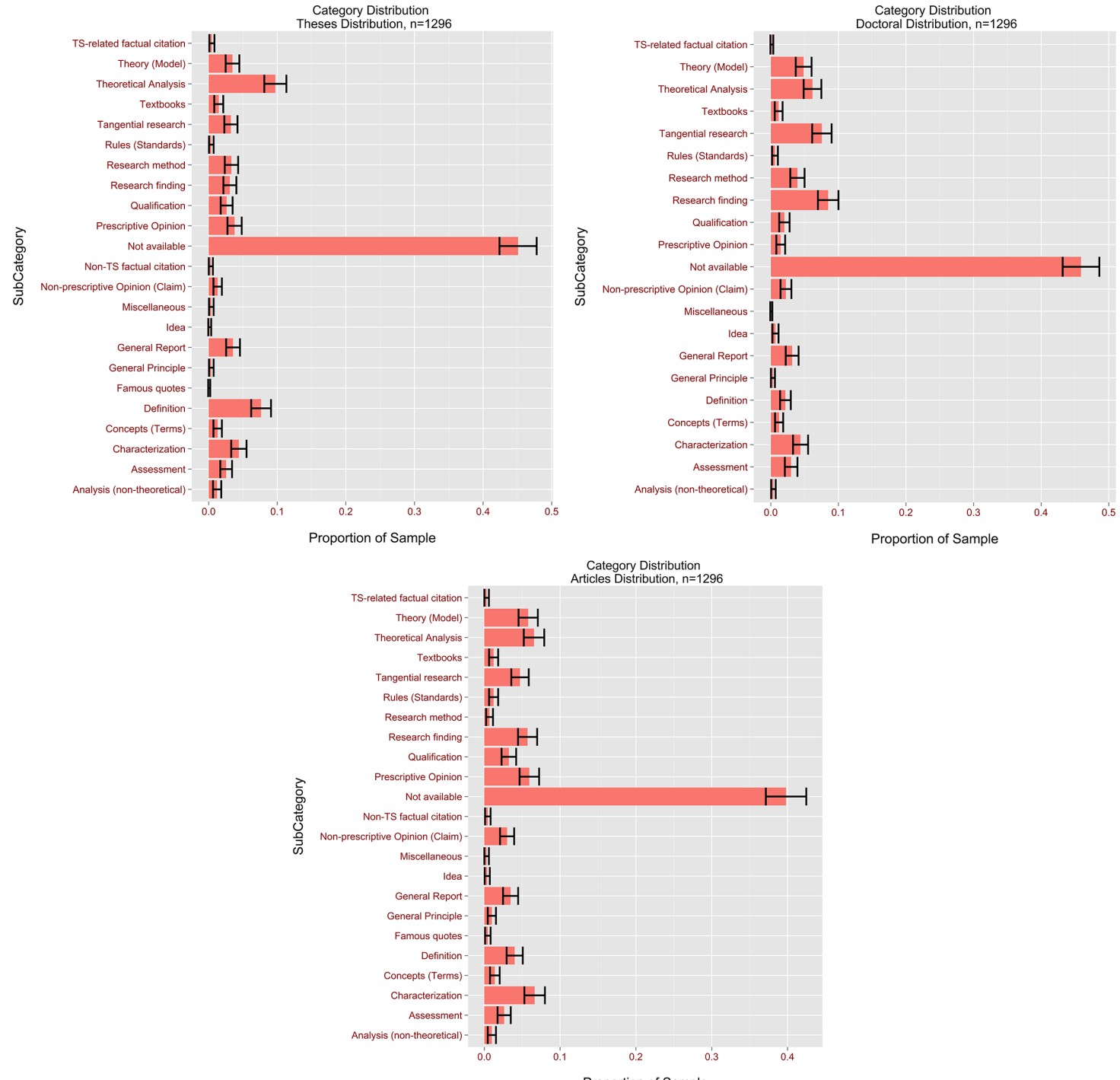

**Figure 8 Proportions of all citation types for MA theses, dissertations and papers.**

 

was [46.2%; 50.6%] for Chinese against [36.6%; 41.0%] for English. The differences in proportions suggest that English references listed in bibliographies are more likely to have corresponding in-text citations than their Chinese counterparts, which would indicate that CIS authors are more indirectly influenced by their Chinese than their Western colleagues.

The analysis also revealed that there was a significant difference in the Theoretical Analysis citation type between Chinese (95% CI [4.1–6.0%]) and English (95% CI [8.5–11.2%]) (Fig. 7). This would appear to indicate that, in cases of theoretical discussions, CIS authors were roughly twice as likely to cite English authors as their Chinese colleagues. In addition, it was found that Prescriptive Opinions (95% CI: Chinese [3.7–5.6%] vs. English 2.1–3.6%) and Textbook citations (95% CI: Chinese [1.6–2.9%] vs. English [0.1–0.5%]) were less common in English than in Chinese citations, while Research Findings were more common (95% CI: Chinese [3.1–4.8%] vs. English [6.5–8.8%]). While advice from all practising professionals is highly valued in the global interpreting community, these findings suggest that the opinions of Chinese interpreters carry more weight with CIS authors than those of their Western counterparts. Textbooks are usually seen as repositories of established fact rather than sources of cutting-edge ideas. The fact that Chinese textbooks are cited more often than Western ones highlights that CIS researchers frequently turn to them for well-established facts.

The most frequently occurring category of Chinese citation besides N/A was Characterization (95% CI [4.6; 6.6%]). Given that experience-based, intuitive writing was dominant in CIS' early developmental stage, and that practising interpreters often resorted to summarising the features of a phenomenon or idea rather than theorising or providing empirical support, its popularity is understandable. Conversely, the most frequent category in English citations after N/A was Theoretical Analysis (95% CI [8.5–11.2%]), which suggests that CIS authors were influenced by the theoretical work of Western authors.

From this point forward the citation distribution in each document type was examined, to spotlight the differences in how different sub-groups of CIS authors make bibliographic references. Citation practice differed widely across the document classes. In MA theses statistically significant differences between Chinese and English references were found for the following citation types: N/A (95% CI [47.1–54.7%] for Chinese vs. 35.6–43.2% for English), Tangential Research (95% CI [3.7–7.1%] for Chinese vs. 0.3–1.9% for English), Theoretical Analysis (95% CI [2.1–5.0%] for Chinese vs. 12.9–18.5% for English) and Textbooks (95% CI [1.5–4.0%] for Chinese vs. 0–0.5% for English). The finding for N/A tallies with the earlier result for all citations. As for the Tangential Research type, the result suggests that Chinese authors' research is more frequently mentioned in passing without specific reference to its contents than is the case for Western authors. One might wonder if some thesis authors cite their compatriots as a way of paying tribute rather than because their works inspire or influence them. In addition, this finding is in line with our interpretation of the differences in proportions for the N/A category between Chinese and English references. Of course, there is also the possibility that thesis authors prefer a style of literature review that summarises more than it analyses. It is worth noting that no statistically significant differences were observed between Chinese and English

references for citation types generally associated with empirical research, such as Research Methodology or Findings.

It was also revealed that Theoretical Analysis was the second most popular type of all thesis citations, though English authors were far more frequently referred to than Chinese for this category. Its popularity across both Chinese and English strongly suggests that theoretical research is influential on Chinese thesis authors. It was somewhat surprising to find that citing someone for their Definitions was a moderately popular citation type (English: 7.9–12.5%; Chinese: 3.4–6.8%)—one might have expected researchers mainly to cite others for their research findings or theories, rather than regularly resorting to them for the definitions of certain terms. Close examination of the contexts in which this type of citation occurs revealed that a significant number of MA students reviewed the history of interpreting at the beginning of their theses, citing Western researchers to define various types of interpreting and clarify the differences between it and translation.

In journal articles and conference proceedings there were statistically significant differences between Chinese and English references for the Prescriptive Opinion (English < Chinese) and Theory citation types (English > Chinese). This finding suggests that in developing their work CIS article authors were more likely to turn to Western scholars for theories and models and to their Chinese colleagues for intuitive understanding of interpreting. It is worth remembering here that the preference for citing Chinese colleagues for Prescriptive Opinion was also observed in theses. It is understandable that Western scholars are more often cited for Theory, because the first generation of Chinese SI trainers received their education in Brussels—theories such as the Interpretive Theory of Translation and the Effort Models have served as the foundations for many a Chinese author's research. It should be noted here that unlike the earlier findings, no statistically significant difference was observed in N/A between English and Chinese references in research papers[9]; this might be explained by the fact that the comparison was made on a smaller sample for papers (the average number of citations in papers was 10, compared to 43 for theses and 278 for dissertations).

In doctoral dissertations the following citation types yielded statistically significant differences between Chinese and English references: N/A (Chinese > English), Research Findings (English > English), Concepts (English > Chinese), and Theory (Chinese > English). As in the dissertation category, there were statistically fewer corresponding in-text citations for Chinese references than for English ones, which is consistent with all previously discussed findings in the overall, article and thesis categories. It should be noted that the proportion of Research Findings was higher here than in any other document type examined. This suggests that PhD students particularly rely on predecessors' empirical findings to shape their own work.

In sum, CIS researchers displayed different citation behaviour across languages and document classes, and no particular citation type yielded significant differences between Chinese and English references across all three document classes. However, the N/A citation type was more frequently used in the Chinese references of theses and doctoral dissertations than in the English ones, while the opposite was the case for Theory. Other

[9] The statistic for a two-sample $z$ test was 0.79, smaller than the critical value of 1.96. There was therefore no evidence to suggest a statistically significant difference between the proportions of English and Chinese N/A citations.

than N/A, no citation type occurred more than 20% of the time across languages and document classes, suggesting that research is cited in diverse ways in the CIS community and no particular citation type is used more than any other.

## Study 3

### Research question

**What different effects does the choice of empirical vs. theoretical research have on the use of citations in the three document types?**

Citations illustrate a dynamic relationship between source and target authors; identifying whether a document is being cited for its methods, concepts or theories illustrates how researchers interact with and influence one another. As observed by *Garfield (1979)*, a comprehensive survey of citation types could provide useful information on the structure and evolution of a science. When a source author is cited for his concepts, ideas or opinions, he is typically engaged in theoretical research, while citations of methodology and findings are typically taken from empirical research (*Gile, 2006*). An examination of the shares of citations that relate to empirical vs. theoretical research would shed light on the relative influences that the two methods have in the three categories of CIS publications.

### Research methodology

The analysis for this research question proceeds from the assumption described in section 'Research question' that certain citation types (Research Methodology and Finding) are typically associated with empirical research, while others (Concepts, Ideas and Opinions) are linked to theoretical studies.

It was expected that doctoral dissertations might contain a greater proportion of empirical citations than MA theses and research papers. Furthermore, it was predicted that citations relating to theoretical research would be most frequently found in papers, followed by theses and dissertations. These predicted outcomes were based on the fact that 80% of dissertations were empirical (compared with 50% of theses and 20% of papers), indicating that empirical methodologies were the preferred research approach among doctoral students, whereas paper authors, who were mostly established academics, preferred theoretical research.

First it was necessary to determine how many data points would be required to detect a 10% difference between two proportions at the 0.05 significance level. Given that there is reason to believe that the differences in the proportions of empirical and theoretical citations are directional, a power calculation for a one-sided two-sample proportion test was conducted to determine the minimum number of citations that would be needed as samples from each of the three categories of CIS publications to guarantee that the test had sufficient statistical power. Since there was found to be no relationship between the hypotheses in Questions 2 and 3, re-using the same samples was not an issue. The minimum sample size for Question 2 was 648 for every possible combination of language and document type, whereas the same number for Questions 3 was 755 for each category of document. The actual sample of 1,296 (combining the sampled Chinese and English

citations in each document type) easily exceeded the necessary minimum for Question 5, further increasing the power without increasing a type one error. Once all the required citations were labelled, a two-proportion $z$-test was performed, yielding $p$-values. The $z$-test examined whether the proportions of citations associated with theoretical research were equal to those associated with empirical.

## Results and discussions

A two-proportion one-sided $z$-test was used to evaluate whether the proportion of empirical citations was greater than that of non-empirical. The test yielded the following $p$-values for the four tests ($p$-values were rounded to 3 digits): the comparison between the proportion of Research Methods and Findings cited by articles and MA theses (empirical citations) was not significant ($p = 0.532$). However, the comparison between doctoral dissertations and theses was significant ($p < 0.001$). The proportions for citations of Research Methods and Findings in theses, articles and doctoral dissertations were 0.064, 0.065 and 0.124 respectively.

(1) Articles > theses IS NOT significant, $p = 0.532$

(2) Doctoral > theses IS significant, $p < 0.001$

It was observed that Research Methods and Findings were cited more in doctoral dissertations than in theses, therefore the null hypothesis that the reverse would apply could be rejected; but there was little evidence to support the same claim for theses as compared to journal articles, therefore that null hypothesis could not be rejected. However, given that the $p$-value in hypothesis (2) was significant, it can be stated with confidence that doctoral dissertations cited research methods and findings more than papers. The data suggest the following relations:

**Research methods and findings:** dissertations > (theses ? journals)[10]

It was observed that theses used fewer citations of Concepts, Ideas and Opinions (theoretical citations) than papers, but there was scant evidence to suggest that doctoral dissertations used fewer such citations than theses. The overall proportions for these citations in the three types of publication were 0.094, 0.086 and 0.133.

(3) Theses > doctoral IS NOT significant, $p = 0.246$

(4) Articles > theses IS significant, $p < 0.001$

However, the present analysis demonstrates that the differences between the two categories are significant, which is in line with the idea that theses use fewer concepts:

**Concepts, Ideas, and Opinions:** (dissertations ? theses) < journals

Because questions (1) and (3) compare MA theses and research papers, and questions (2) and (4) compare doctoral dissertations and MA theses, and because there is no new random sample for each test, the tests are correlated. To control the familywise error rate, we implemented the Bonferroni-Holm correction method, a sequential method where the $p$-values do not have to be adjusted but can, rather, be compared to a different significance level to keep the familywise error rate for all of the tests at 0.05. Without such correction, there would be an increased chance of rejecting correct null hypotheses, thus increasing Type-I errors.

[10] 'Theses? journals' means that the null hypothesis that theses have fewer research methodology citations than articles cannot be rejected, therefore the author cannot comment on any relationship that might exist between theses and journals in this specific case. Unlike a two-sided test, where the null hypothesis is equality, the null hypothesis for our one-sided test is less than or equal to zero.

After the correction, even the two smallest $p$-values remained significant when compared to the new cut-offs:

(2) Research Methods and Findings: doctoral > theses IS significant at 0.0125, $p$-value: 7.75e−08

(4) Concepts, Ideas, and Opinions: articles > theses IS significant at 0.0167, $p$-value: 0.000639

This strongly suggests that the previous findings—that dissertations cite more Research Methods and Findings than theses, and that dissertations and theses cite fewer Concepts, Ideas and Opinions than papers—still hold true even after adjusting for familywise errors. In other words, the choice of empirical research has a greater influence on how doctoral scholars cite papers than it does on thesis and paper authors; and theoretical research has a more significant impact on how paper authors make citations than it does on MA and doctoral authors. This finding is interesting in that it indicates that data-driven research is more popular among MA and doctoral students, whereas theoretical research is favoured by established academics. If MA and doctoral students represent the future of CIS, one can expect empirical research to continue to expand its sphere of influence in the foreseeable future.

## CONCLUSION

The authors hope that this scientometric survey has demonstrated the merits of blending quantitative with qualitative analysis to paint a panorama of Chinese Interpreting Studies. Citation data was used to measure the progress of CIS: the field is a dynamic one with new papers being constantly cited, though a few influential older papers have withstood the process of 'obliteration'. Citation sampling and labelling were employed to describe how scholars exhibit different citation patterns across languages and document types, and how authors' choice of empirical or theoretical research variously influences journal articles, MA theses and doctoral dissertations.

Thanks to its comprehensive collection of data, the study authors did not face the issues of sample size typically associated with quantitative analyses of interpreting studies (IS) in the past. In the West only a few hundred individuals are dedicated to interpreting research; by contrast, no fewer than 3,500 Chinese scholars are documented in this study's database: this wealth of available data made it possible to adopt some of the latest statistical techniques to assess the evolution of CIS. The qualitative elements of the study served to spotlight unique patterns of behaviour exhibited by its researchers when citing their predecessors.

The focus of this paper being IS, no attempt was made to explore translation-related publications. Given that translation studies (TS) has a longer history and many more participants than IS in China, an interesting future line of inquiry would be to research its themes and patterns, so as to offer a balanced view of how the whole discipline of translation and interpreting studies has developed and is developing. According to *Franco Aixelá (2013)*, it takes at least five years for Western TS publications to receive any citations, in sharp contrast to the 1–2-year rule for Chinese papers. It is reasonable to

predict that Chinese TS, with its strong focus on classic authors and their theories, may need much more time to gain any traction. However, as is the case for IS, with its increasing influence from neighbouring disciplines such as psychology and linguistics, the time lag between citing and cited papers in TS is greatly shortened as time goes by.

Also, with the ever-increasing level of academic exchange between East and West, a comparative study of the two would provide valuable insights to policy-makers charged with shaping the future direction of academic research. *Chesterman (1998)* observed that one of the most important trends in TIS has been the shift from theoretical to empirical research. *Pym (2009)* further argued that the regurgitation of 'authoritative' insights without empirical testing does not help advance the field. It would be interesting to seek out empirical evidence to corroborate Chesterman's observations regarding the trends in Western TIS and to investigate whether empirical studies have a tangible effect on journal articles, theses and doctoral dissertations in the field. If the answer to that question turned out to be negative, the design of curricula for training future TIS researchers might need to be reconsidered, and in particular their empirical components reinforced. On the Chinese front, Wang and Mu found (*2009*) that though the number of publications was on the rise, many were no more than speculative and personal reflections regarding training. Thanks to its much more comprehensive and recent data-set, this study has found that the current CIS landscape is a complex and rapidly changing one: while journal articles and conference proceedings are still dominated by theoretical analysis, doctoral studies have been heavily influenced by empirical research.

Interpreting is a profession whose actions can have real-world consequences in diplomatic, military, commercial, and judicial fields, among many others. For this reason alone, the ideas and theories currently in use need to be tested, adjusted and improved in the real world so that aspiring interpreters may hone their skills with the maximum of efficiency and efficacy. The findings of this study have demonstrated that Chinese Interpreting Studies has come a long way from its infancy. While much more work undoubtedly needs to be done to improve the overall quality of CIS research, it is reassuring to have learned that scholars in the field have begun to appreciate the importance of empiricism and that as a result the general trend of the discipline is moving in the right direction.

## ACKNOWLEDGEMENT

The authors wish to thank Ewan Parkinson for patiently reviewing multiple versions of this paper and for generously providing detailed suggestions on improving its quality.

### Funding

The authors did not receive external funding for this research.

### Competing Interests

The authors declare there are no competing interests.

## Author Contributions

- Ziyun Xu conceived and designed the experiments, performed the experiments, analyzed the data, contributed reagents/materials/analysis tools, wrote the paper, prepared figures and/or tables, reviewed drafts of the paper.
- Leonid Pekelis conceived and designed the experiments, performed the experiments, analyzed the data, contributed reagents/materials/analysis tools, reviewed drafts of the paper.

## Data Availability

https://github.com/danielxu85/scientometrics.

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
