# Peer review of "Chinese Interpreting Studies: a data-driven analysis of a dynamic field of enquiry"

_PeerJ, doi:10.7717/peerj.1249_

## Round 0.1 · original submission · Major Revisions

The research work performed in this Manuscript is, without any doubt, interesting and worth publishing. The data set the authors have created is really impressive, and it was analysed in great detail.

I’m nevertheless afraid that the Manuscript cannot be published in its current version. As the three Referees have pointed out, the text is extremely long - more than 80 pages! This firstly was a barrier for securing three Referees (and, about that, I have to thank them for the great work they have performed). But secondly, it will be a serious problem for the published paper: sadly, nowadays very few researchers have time to read through 80 pages, even if the article is really interesting! Thus, the length of the paper will be an important barrier for creating future impact.

It is really important that the authors reduce the length of the paper to, at least, 40 pages - even though 30 would be a much better target. This is not as difficult as it may appear. The Appendix is probably not essential, and can be deleted (maybe including in the main text a couple of sentences about the rationale for using Varying Coefficient Models). Many figures can be joined together: for instance, Figs 1 - 4 can be one single figure, composed of four panels. Also, some figures have a lot of white (empty) space, as for instance Fig. 26. Finally, I agree with Referee 2 that repeating twice the research questions is not very efficient: authors may try to restructure the text to make it more compact. After all this, authors may try to make the text more synthetic, avoiding sentences that are just a repetition of the content of the images - see, for instance, the following one, that does not yield any added value with respect of the two referenced figures: “While the average for the former hovered just above 80 from 1997 to 610 2013 (see Figure 17), the figure for the latter increased from around 120 in 2004 to nearly 150 in 2008 611 and then declined sharply to around ten in 2011 (see Figure 18).”

Beyond that, I have two minor comments:

1. I agree with Referee 3 that the reproducibility of the analysis is an important issue. Would it be possible to publish the data set, as a part of the paper, or even by uploading it in the authors website? Please notice that this will also improve the value of the paper in the long term - and will help attracting more citations!

2. All texts inside figures are really small. Especially is multiple images are joined together, it is necessary to significantly increase the font size.

Reviewer 1 ·

Basic reporting

.

Experimental design

.

Validity of the findings

.

Additional comments

I'm very impressed by the job the authors have done to create a citation database from scratch and to perform very detailed statistical analyses on several topics, but I will not go further in the review process until the manscript is re-processed in a proper journal-article format. The current format is more similar to a ''project work'' with a series of question/answer/graphs blocks scattered throughout the 80 pdf pages (a journal article cannot be so long). Please, before re-submitting it, give the paper a coherent structure Introduction, Previous Research and Research Questions /Materials and Methods / Results / Discussion / Bibliographic References.

·

Basic reporting

The article presents very interesting data concerning Chinese Interpreting Studies. It certainly adds to the growing body of literature concerning CIS, while also offering a new perspective --that of scientometrics. While the data is original and of great interest, the article is a bit difficult to read due to its structure. Perhaps the authors could consider grouping all the information referring to each research question and presenting it after the question. This would help avoiding the repetition of each research question when the authors present the methodology and the results and discussion, and would also help the reader better contextualise the data. The article is also very long, especially if considering that most research papers are not longer than 30-40 pages. Perhaps the authors could consider selecting only the most interesting research questions and focusing on only two-three (and leaving the others for another article).

Experimental design

The research is based on a solid methodology and has been conducted rigorously and to a high technical standard. Moreover, it reflects the enormous amount of research work developed by the authors.

Validity of the findings

Perhaps the conclusions could be a bit broader. While the article presents a lot of interesting findings which are all very relevant to understand how CIS are developing in China, the conclusions only present a summary of the kind of research that has been presented, but no implications or broader conclusions are mentioned. After such huge effort as is the case in a scientometric survey, the reader would expect conclusions that also related the findings to the reality of CIS.

Reviewer 3 ·

Basic reporting

The article is well written and organized. Also, it is quite long, some section should be reduced or replaced. For example, the appendix is not necessary.

Experimental design

The research question are clearly defined. But, the main drawback is the data collection, because it cannot be reproduced by the research community. Authors should use Isi Web of Science or Scopus, or at least, provided the raw data or database as a supplementary material

Validity of the findings

The results and findings are adequates.

---

## Round 0.2 · Minor Revisions

As clearly indicated by the two Referees, with who I agree, the manuscript has substantially improved from the initial version, and it is now almost ready for publication. I would nevertheless ask the authors to address the two small comments raised by the first Referee, as they are important respectively to make the paper more complete and more technically correct.

Reviewer 1 ·

Basic reporting

The article is well-researched and fairly well written, thus appearing more suitable for publication in an international journal than the previous version. Its main contribution, however, is not so much in the advanced statistical modelling – which is like using a bazooka to kill a fly given the shaky epistemological background of citation analysis – but in the provision of original, otherwise inaccessible data on a relevant subset of the Chinese scholarly universe.Some methodological and content-related issues, however, need to be pointed out.

Experimental design

Two remarks:

1) I understand the necessity to manually put together publications and citations metadata given the lack of coverage in international citation databases, but why jump directly to sophisticated inferential statistics without presenting at least a table of descriptive data on the source items (distribution of publication years, languages, article types, number of authors, most cited journals, authors, papers, etc.)?
2) Is the use of general linear models (fitting CIS citation data), standard confidence intervals and z-tests justified? Did you check for the assumptions required in order for such techniques to yield valid results?

Validity of the findings

“Convenience sampling should not be an issue in this study: it would only be a problem if there were some inherent qualities among those one was able to sample that would not be present in the entire population” (165).
This is a very “convenient” claim: the sample is certainly big enough to allow sound statistical calculations, still it is not random and it is not even assembled following a rational criterion (such as all the articles of the most important journals publishing CIS studies), so the conclusions cannot lightheartedly be extended to the entire population of CIS studies.

Reviewer 3 ·

Basic reporting

Authors have done the majority of the suggestion, therefore, I recommend to accept the paper.

Experimental design

Following my previous suggestion, the authors have made available the datasets.

Validity of the findings

The conclusion have been broadned

---

## Round 0.3 · accepted · Accept

As the authors have answered to the last few comments of the referees in a satisfactory way, the manuscript is now ready for publication.